# On chaotic dynamics in transcription factors and the associated effects in differential gene regulation

Mathias L. Heltberg[1], Sandeep Krishna[2] & Mogens H. Jensen[1]

The control of proteins by a transcription factor with periodically varying concentration exhibits intriguing dynamical behaviour. Even though it is accepted that transcription factors vary their dynamics in response to different situations, insight into how this affects downstream genes is lacking. Here, we investigate how oscillations and chaotic dynamics in the transcription factor NF-$\kappa$B can affect downstream protein production. We describe how it is possible to control the effective dynamics of the transcription factor by stimulating it with an oscillating ligand. We find that chaotic dynamics modulates gene expression and up-regulates certain families of low-affinity genes, even in the presence of extrinsic and intrinsic noise. Furthermore, this leads to an increase in the production of protein complexes and the efficiency of their assembly. Finally, we show how chaotic dynamics creates a heterogeneous population of cell states, and describe how this can be beneficial in multi-toxic environments.

[1] Niels Bohr Institute, University of Copenhagen, Blegdamsvej 17, DK-2100 Copenhagen, Denmark. [2] Simons Centre for the Study of Living Machines, National Centre for Biological Sciences-TIFR, GKVK Campus, Bellary Road, Bangalore 560065, India. Correspondence and requests for materials should be addressed to S.K. (email: sandeep@ncbs.res.in) or to M.H.J. (email: mhjensen@nbi.dk)

The regulation and control of protein production is a vital element in all living organisms. This process can be highly complicated, involving a large number of steps. However, despite stochastic fluctuations, life is characterised by a high level of organisation indicative of very precise regulation. A thorough understanding of the mechanisms and interactions that maintain the precision of regulation is absent, but the prospect of discerning and ultimately controlling the production of specific proteins is one of the great goals in the field of systems biology.

Control of transcription is a ubiquitous means of regulating gene expression, but it has only recently been appreciated that transcription factor dynamics might be important for gene regulation. For instance, oscillations have been observed in key transcriptional factors, such as the p53 tumour suppressor or NF-κB, which regulates numerous genes involved in immune response[1–7]. Debate continues about the functional role, if any, of these oscillations, but it is clear that altering the dynamics of these transcription factors differentially affects downstream genes[1,2,4,8].

Oscillatory dynamics is the prerequisite for many complex phenomena—and in the present study for the onset of chaotic dynamics. Chaos refers to complex, apparently unpredictable, dynamics that even simple deterministic dynamical systems can produce (see section What is chaos?). A universal way to achieve chaos is by driving a nonlinear oscillator (say the NF-κB system) by an external periodic signal (e.g., by periodically varying a cytokine-like tumor necrosis factor (TNF) that triggers NF-κB oscillations). When the external driving signal has low amplitude oscillations, it can entrain or synchronise the nonlinear oscillator, i.e., if TNF is varied within certain frequency ranges it will force the NF-κB oscillations to occur with the externally imposed frequency[8,9]. As the amplitude of TNF oscillations is increased, the range of frequencies for which it can entrain NF-κB becomes larger—these expanding synchronisation regions of the external amplitude–frequency parameter space are called Arnold tongues[10–12]. Such entrainment/synchronisation[13] has been observed in many different physical systems, from fluids[14] to quantum mechanical devices[15,16], and now also in biological processes, such as cell cycles[17–19], and gene regulatory dynamics in synthetic populations[20]. The dynamics gets even more complex as the amplitude of the external driving signal increases further. First, Arnold tongues start overlapping, which means the nonlinear oscillator can exist in more than one entrained state with different frequencies (termed modes), and even small amounts of intrinsic or extrinsic noise can cause it to hop between these modes. Such mode-hopping has been observed in the NF-κB system when driven by a periodically varying TNF signal of sufficiently high amplitude[21]. When the external amplitude is increased even further, then chaotic dynamics is predicted[11,12].

In this paper, we study the possible implications of oscillatory and chaotic dynamics of a transcription factor, such as NF-κB, on the downstream genes it controls. We compare the expression of genes with different affinities to the transcription factor, and show that chaotic dynamics has differential effects on genes with different affinites. This can be exploited, for instance, to up-regulate certain proteins, or specific protein complexes. We also show how chaotic dynamics can generate heterogeneity in a cell population that can provide a selective advantage in multi-toxic environments. Our work provides a theoretical framework to study the effects of dynamically varying transcription factors, and we believe it constitutes one of the first investigations into how chaotic dynamics might influence genetic regulation in living cells.

## Results

**The model**. Our investigation starts with a model of the transcription factor NF-κB that is known to exhibit oscillatory dynamics[3,9,22]. A schematic version of this is found in Fig. 1a and a full description is presented in the Supplementary Note 1. In this deliberately simplified model, the oscillations arise from a single negative feedback loop between NF-κB and its inhibitor IkBα, and can be triggered by TNF via the activation of the IkB kinase (IKK). We then allow TNF to oscillate. This system exhibits Arnold tongues (shown schematically in Fig. 1c), which are regions of parameter space where the NF-κB oscillation is entrained to the external TNF oscillation[9], i.e., it locks on to the external signal's frequency and phase. Outside the Arnold tongues there is no synchronisation. It is straightforward to add intrinsic noise to this system by explicitly modelling the randomness in binding/unbinding of proteins, phosphorylation, as well as transcriptional and translation processes using the Gillespie algorithm[23] (see Supplementary Note 2 and Supplementary Figure 1C for details). Figure 1c right panel shows that this system exhibits single-mode oscillations (for low amplitude TNF oscillations), mode-hopping (intermediate amplitude) and chaos (high amplitude) in this system, as was first noted in ref. [21]. Note that changes in a single parameter are sufficient to obtain all these different dynamics.

To the above model, we now add genes that are regulated by NF-κB, following the approach of Mengel et al. [4]. We assume that NF-κB can bind to an enhancer or operator region, and can form complexes to bind the RNA polymerase, with different affinity, depending on the gene (schematically shown in Fig. 1d). We describe the transcription and translation of each gene, labelled $i = 1, 2, 3,...$, using the differential equations:

$$\dot{m}_i = \gamma_i \frac{N^{h_i}}{N^{h_i} + K_i^{h_i}} - \delta_i m_i, \quad (1)$$

$$\dot{P}_i = \Gamma_i m_i - \Delta_i P_i. \quad (2)$$

Here, the $m_i$ represent the mRNA level transcribed from gene $i$, and $P_i$ represents the concentration of proteins produced from the correspnding mRNA. The first term in the equation for the mRNA is known as a Hill function; the canonical way to describe the protein production for genes governed by transcription factors where each gene has a specific Hill coefficient and effective affinity[4,24–27].

The effective affinity $K_i$ is a parameter that combines the strength of binding of the transcription factor to the operator/enhancer region, the strength of binding of RNA polymerase to the promoter and transcription factor, as well as the effect of DNA looping that may be needed to bring the enhancer/operator close to the promoter region. Operationally, $K_i$ sets the concentration of NF-κB that results in 50% of maximal gene expression enhancement. The Hill coefficient $h_i$ is a measure of the cooperativity of the transcription factor at that gene. A thorough description of this is presented in Supplementary Note 3, and representations of the sigmoidally shaped curves are shown in Supplementary Figure 1. $\gamma_i$ and $\Gamma_i$ are the maximal transcription and translation rates for the gene, while $\delta_i$ and $\Delta_i$ are inversely proportional to the half-lives of the mRNA and protein, respectively. While all these parameters affect the behaviour of genes described by these equations, the affinity $K_i$ is particularly important. In particular, as we will demonstrate in subsequent sections, high-affinity genes (HAGs) with low $K_i$ behave quite differently from low-affinity genes (LAGs) with high $K_i$. In Fig. 1e the values of the Hill function $N^h/(K^h + N^h)$ as NF-

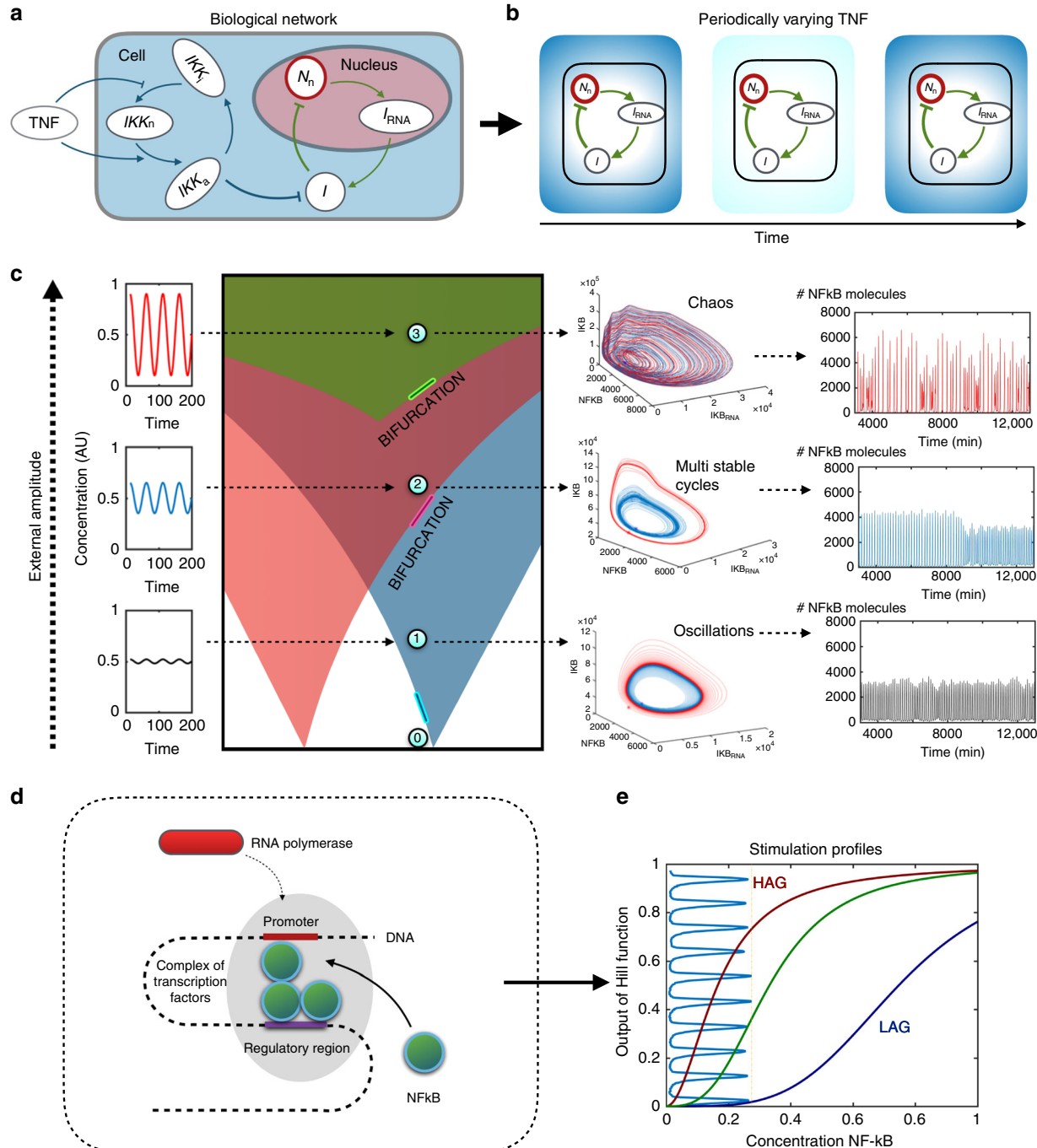

**Fig. 1** Dynamics from coupled oscillators and emergence of chaos. **a** Schematic picture of a simplified NF-$\kappa$B network with a single negative feedback loop which can generate oscillations. **b** Schematic picture of oscillations in the external TNF concentration, represented by the changing shade of blue. **c** Dynamics that emerges when the NF-kB system is driven by a periodic TNF signal. The left panel shows schematically that there are Arnold tongues, triangular regions of the TNF amplitude-period parameter space where NF-kB oscillations can be synchronised to the TNF signal. Outside the Arnold tongues, e.g. point 0, there is no synchronisation. As TNF amplitude increase the Arnold tongues start overlapping and the behaviour becomes more complex. Keeping the TNF period fixed (here we used $T = 50$ min), as we increase the the TNF amplitude we enter three distinct states: Point 1: A single Arnold tongue, only allowing one oscillation state. Point 2: Overlap of Arnold tongues, allowing two stable oscillation states. The presence of noise can cause transitions (mode-hopping). Point 3: Chaotic dynamics, with apparently unpredictable trajectories. The trajectories corresponding to these points are shown in in the middle panels. Red and blue trajectories correspond to two different initial conditions in a deterministic simulation. The rightmost panels show the dynamics of NF-$\kappa$B vs. time in stochastic simulations where intrinsic noise is implemented using the Gillespie algorithm. **d** Schematic figure of the polymerase binding for genes that have NF-$\kappa$B (green spheres) as a transcription factor. **e** Profile of the Hill function in Eq. (2) for different values of affinity and cooperativity. Red: $h = 2$ and $K = 1.0$ (HAG). Purple: $h = 4$ and $K = 4.5$ (LAG). Green: Example of intermediate values with $h = 3$ and $K = 2.0$ (MAG). Vertically, light blue: a representative NF-$\kappa$B oscillation

$\kappa$B level $N$ varies are shown, along with a typical single-mode NF-$\kappa$B oscillation. It is clear from the figure that the same NF-$\kappa$B oscillation would be expected to excit HAGs and LAGs to different levels.

**What is chaos?** When we speak of chaos, we refer to deterministic chaos. Deterministic means that if one knows the initial state of the system exactly, then the dynamical trajectory will be the same every time it is initiated in that state. However, any two initial conditions infinitesimally apart will have exponentially diverging trajectories as time proceeds making it practically impossible to predict the future dynamics—hence chaos[28–31]. It is important to note that the unpredictability of chaos does not arise from stochasticity—the latter refers to a non-deterministic system with noise. Noise is observed in most real-world systems and can often result in very different dynamics than the deterministic version of the same system. For example, noise can cause transitions between different states which would never occur if the system were deterministic. Thus, both deterministically chaotic and noisy systems exhibit unpredictability of their future trajectories, but for very different underlying reasons.

**Chaos enhances LAGs.** We simulate our model of the NF-$\kappa$B system, with periodically varying TNF and intrinsic noise, along with downstream genes with different affinities and cooperativities. We then measure the average protein concentration associated with each gene over timescales much longer than the half-lives of the mRNA and proteins. This long-term average is the simplest measure of the effect of NF-$\kappa$B oscillations on gene expression. As shown in Fig. 2, we find that as TNF amplitude increases, we obtain very different behaviour for HAGs, LAGs and genes with intermediate affinity (MAGs). As described above and in Fig. 1, as TNF amplitude is increased, keeping its frequency fixed, the NF-$\kappa$B dynamics is first a single-mode oscillation (point 1 in Fig. 1c), then exhibits mode-hopping (point 2 in Fig. 1c) and finally chaos (point 3 in Fig. 1c) for high amplitude TNF. The ranges of TNF amplitude which exhibit these three qualitatively different dynamics are indicated in Fig. 2a–c.

The chaotic regime shows the differential behaviour of the different genes most clearly. The HAG has a linearly decreasing average protein level as TNF amplitude is increased, while the LAG shows exactly the opposite. The MAG exhibits much less variation with TNF amplitude. It is interesting that genes under control of NF-$\kappa$B can thus be designed to have increasing, decreasing, as well as relatively flat response to variation of a single parameter. The increasing (decreasing) trend that is seen for LAGs (HAGs) within the chaotic regime is also seen across the entire range of TNF amplitudes, going from single-mode oscillations through mode-hopping to chaos. However, within the first two regimes the response is relatively flat with major change happening only near the transition between regimes. Overall, we see that both HAGs and LAGs could exhibit fold-changes on the order of two-fold, which we believe should be observable in experiments, while MAGs could lie within experimental error and thus appear effectively unresponsive to TNF amplitude.

A mathematical analysis of this behaviour provides some intuition to understand why HAGs and LAGs respond so differently: The long-term average protein level is essentially proportional to the average of the Hill function over the same long timescale: $\langle P \rangle \sim \left\langle \frac{N^h}{K^h + N^h} \right\rangle$. For HAGs, $K$ is small, and to lowest order in $K/N$, $\langle P \rangle \sim 1 - K^h \langle \frac{1}{N^h} \rangle$. In contrast, for LAGs, $K$ is large, and $\langle P \rangle \sim \frac{\langle N^h \rangle}{K^h}$ (see Supplementary Note 4 for further details). The averages $\langle N^h \rangle$ and $\langle 1/N^h \rangle$ depend on the probability distribution of NF-$\kappa$B values over a long time series. This distribution is typically unimodal, but is asymmetric and has a long right tail (see Supplementary Figure 2K–L). Now $\langle N^h \rangle$ is largely dominated by this right tail, especially for large $h$. Thus, if the right tail of this distribution became more prominent as TNF amplitude was increased, we would expect $\langle N^h \rangle$ to increase and this would explain why LAGs show an increasing average protein level, while $\langle 1/N^h \rangle$ in contrast is dominated by the other end of the probability distribution, i.e., very low values of $N$. Thus, if the probability of NF-$\kappa$B spending time at low concentrations increased with TNF amplitude, then $\langle 1/N^h \rangle$ would increase, and the average protein level of LAGs would decrease. Supplementary Figure 2K–L shows that this is indeed what happens to the probability distribution of NF-$\kappa$B as TNF amplitude is increased—both within the chaotic regime, as well as across single-mode oscillations, mode-hopping and chaos. Thus, we conclude that the differential control of HAGs vs. LAGs is directly caused by the broadening of the range of NF-$\kappa$B levels as one goes deeper into the chaotic regime. The increase of peak NF-$\kappa$B levels and the decrease of minimum NF-$\kappa$B levels are both necessary for such differential control.

**Robustness to variations in parameters and noise.** We tested our central result from the previous section at other TNF frequencies (see the heatmaps in Fig. 2j–l and Supplementary Figure 2A–F) and, for TNF time period in the range 30–120 min, we found the same trends in average protein levels, as a function of TNF amplitude.

Since biological systems are often characterised by large fluctuations and much noise, we also varied the level of intrinsic noise in the NF-$\kappa$B system by varying the effective volume of the system. Decreasing the volume leads to larger fluctuations, but as shown in Fig. 2d–f, the average protein levels are quite robust to such increases of intrinsic noise. The mode-hopping region is of specific interest to changes in noise, since these affect the rate at which the system jumps from one entrained state to another[32]. The chaotic regime, in contrast, already exhibits many hallmarks of randomness even in the absence of noise, so adding noise does not affect the behaviour much.

Next, we also wanted to include extrinsic noise into the variation of TNF. In experimental procedures, as well as in vivo, it is of course very likely that there will be considerable stochasticity in the TNF signal. Could such fluctuations mask the differential control of genes, especially in the chaotic regime? We added Langevin noise to the periodic TNF waveform at a sufficiently high level to smear out the predominant frequency in a Fourier spectrum of the noisy waveform (see Supplementary Figure 2G-H). We found that this did not affect our results— NF-$\kappa$B still showed the same transition from single-mode oscillation to mode-hopping to chaos as TNF amplitude was increasing, and HAGs and LAGs showed the same opposite trends in average protein level as in the absence of TNF noise (Fig. 2g–i). As with intrinsic noise, the extrinsic noise had most effect in the mode-hopping regime and minimal effect in the chaotic regime.

Finally, we also found that our results were unchanged when we used non-sinusoidal waveforms for TNF (Supplementary Figure 2I), and when we varied the Hill coefficient (Supplementary Figure 2J). Thus, these results show that the enhancement of LAGs in chaotic dynamics is robust to both internal and external noise, and this effect is a striking feature of chaos in transcription factors for a large set of parameters.

**Chaos increases efficiency in protein complex formation.** In eukaryotic cells, many functions are carried out by complexes of proteins that are constructed from multiple subunits, for

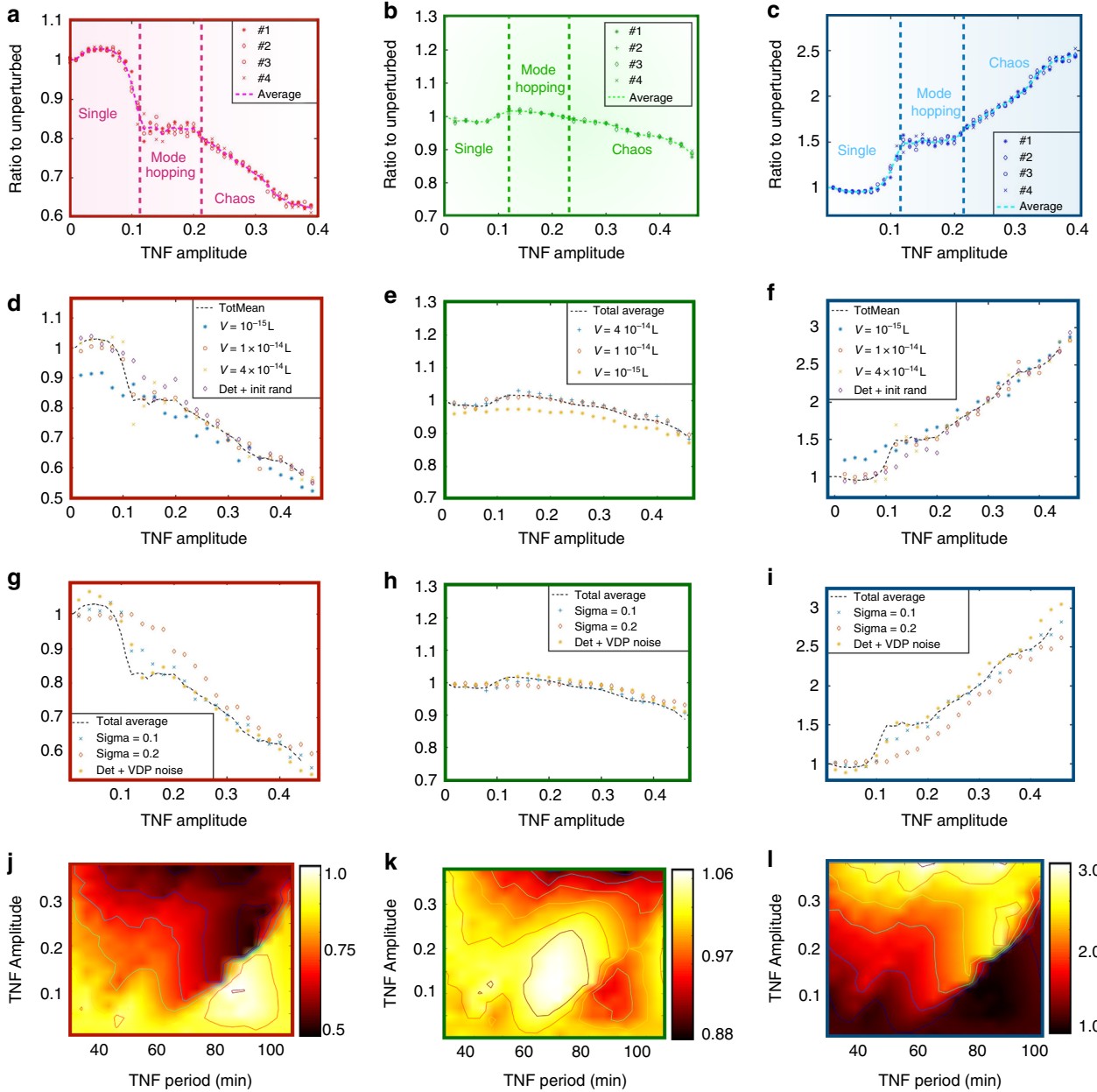

**Fig. 2** Effects of chaos on protein production. **a** The average protein level from an HAG ($K = 4.5$, $h = 4$), for different values of the TNF amplitude. We performed four separate simulations of duration $5*10^5$ min. and show their individual means and the mean of these four combined. We used $V = 2*10^{-14}L$ and $T = 50$ min. Results for other periods are found in SI. **b** Same as **a** but for MAG ($K = 2$, $h = 3$). **c** Same as **a** but for LAG ($K = 1$, $h = 2$). **d** The average production from the HAG for different intrinsic noise levels. We performed simulations of duration $1*10^5$ min each datapoint and used the total mean as calculated in **a**–**c**. Here $T = 50$ min. **e** Same as **d** but for the MAG. **f** Same as **d** but for the LAG. **g** The average production from the HAG for different extrinsic noise levels. We performed simulations of duration $1*10^5$ min each datapoint and used the total mean as calculated in **a** and **b**. We added Langevin noise to the oscillator and used $\dot{r} = r(1 - r)$ and $\dot{\theta} = \nu$, and in the last dataseries (yellow *) we used the Van der Pol oscillator with noise as a perturbation to TNF (see Supplementary Note 4). Here $T = 50$ min. **h** Same as **g** but for the MAG. **i** Same as **g** but for the LAG. **j** The average production from the HAG, for different values of the TNF amplitude and period. The bright colours indicate the maximal average protein levels, while the dark colours correspond to low average protein levels. **k** Same as **j** but for the MAG. **l** Same as **j** but for the LAG

instance haemoglobin, that consists of four subunits from two genes that are located on different chromosomes. A study of the NF-$\kappa$B interactome found that amongst 384 genes that are regulated by NF-$\kappa$B there were 572 protein–protein interactions[33]. While these complexes have not been deeply investigated, we expect at least some will have a functional role. For instance, there seems to be evidence that NF-$\kappa$B controls

autophagy via multiple pathways, including the up-regulation of both Beclin 1 and A20, which interact with each other inhibiting Beclin 1 ubiquitination, and thereby repressing autophagy[34]. Therefore, we tested how the concentration of protein complexes, whose subunits were encoded by NF-$\kappa$B controlled genes, was altered as the NF-$\kappa$B dynamics became chaotic and the LAGs were up regulated.

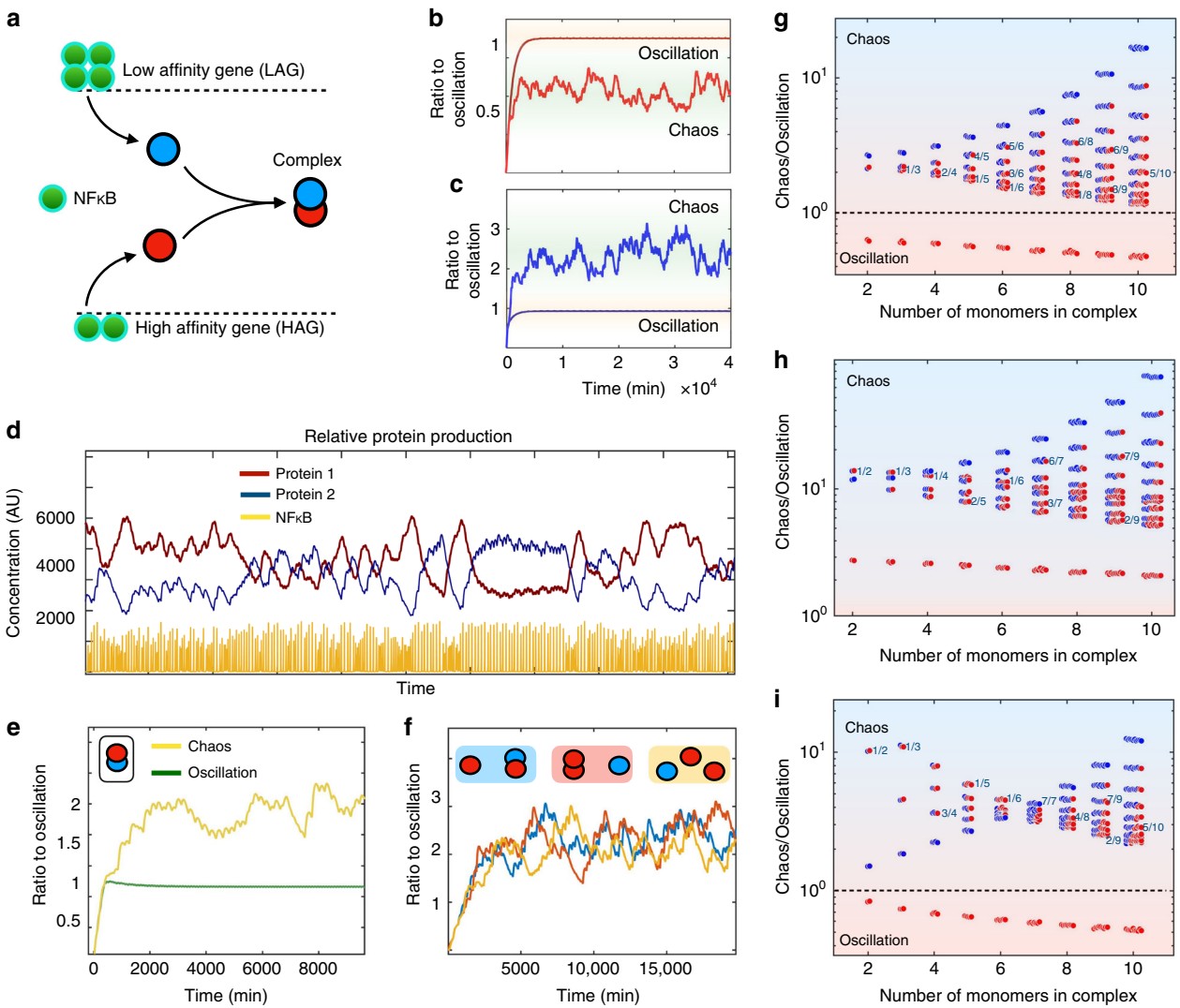

**Fig. 3** Effects of chaos on protein complex formation. **a** Schematic picture of how LAG and HAG encoded proteins may form protein complexes. **b** Protein concentration for HAG ($K = 1$, $h = 2$) with oscillatory and chaotic dynamics, respectively. $T_{TNF} = 50$ min. **c** Protein concentration for LAG with ($K = 4.5$, $h = 4$) oscillatory and chaotic dynamics, respectively. $T_{TNF} = 50$ min. **d** Time series of NF-$\kappa$B (yellow) and the corresponding protein level. Red: Level of HAG protein. Blue: Level of LAG protein (multiplied by 20). $T_{TNF} = 50$ min. **e** Concentration of a heterogenous two-protein complex (shown upper left) with oscillatory and chaotic dynamics respectively. **f** Concentration of a heterogenous three-protein complex in chaotic dynamics depending on the hierarchical assembly (shown above). **g** Relative concentration for complexes of different compositions. The $y$-axis show the concentration in chaos divided by the concentration in the oscillatory regime, and the black line show where these are equal. **h** Relative concentration per NF-$\kappa$B for complexes of different compositions measured by the fraction $\frac{\langle C_{N,n_H} \rangle}{\langle NFkB \rangle}$. Same axis as in **g**. **i** Relative concentration per unused subunits for complexes of different compositions measured by the fraction $\frac{\langle C_{N,n_H} \rangle}{\sum_{i=1}^{N} \langle P_i \rangle}$. Same axis as in **g**. For **g–i**, the exact ratios are found in the tables in Supplementary Note 5

We first consider a complex that consists of two subunits. In this case, the model has the following additional equations, where $P_1$ and $P_2$ represent the concentrations of the two proteins and $C_{2,1}$ the concentration of the complex:

$$\dot{P}_1 = \Gamma_1 m_1 - \lambda_C P_1 P_2 - \Delta_1 P_1, \tag{3}$$

$$\dot{P}_2 = \Gamma_2 m_2 - \lambda_C P_1 P_2 - \Delta_2 P_2, \tag{4}$$

$$\dot{C}_{2,1} = \lambda_C P_1 P_2 - \Delta C_{2,1}. \tag{5}$$

In the following we will, in order to keep things as simple and transparent as possible, keep the values of the parameters $\lambda$ and $\Delta$ fixed even though these could easily differ between complexes. An exploitation of the effects of the entire parameter space will be interesting to pursue in future work, but is beyond the scope of this paper. Obviously if the two subunits are both HAG proteins, the complex has the highest average level in the oscillatory regime, while if it consists of two LAG proteins, the highest average level will be found in the chaotic regime (Fig. 3b, c). However, if the complex is heterogeneous and consists of one HAG and one LAG subunit, as shown schematically in Fig. 3a, the result is not as obvious. Simulating the above equations for a heterogenous complex, we

find a significantly higher level of the complex in the chaotic regime, as seen in Fig. 3e.

We then test larger complexes. The concentration of the protein complex $C_{n,n_H}$ consisting of $n$ subunits, of which $n_H$ are from an HAG and the rest from an LAG, is modelled by

$$\dot{C}_{n,n_H} = \lambda \prod_{i=1}^{n} P_i - \Delta_n C_{n,n_H}. \tag{6}$$

For $n = 3$ and $n_H = 2$, we found that the production was also highest in the chaotic regime and, before moving further, we tested whether the outcome was different if all complexes combined randomly (yellow curve in Fig. 3f), or if there was a hierarchical structure in the assembly (blue and red curves in Fig. 3f). As we see in Fig. 3f the outcome is quite similar, and we could therefore focus on the non-hierarchical assembly of complexes, calculated as shown above. We subsequently tested for $n \in [2{-}10]$ and in each case we tried with all different different combinations of HAG and LAG subunits. Unexpectedly, we find that all heterogeneous complexes exhibit a higher average lvel in the chaotic regime (Fig. 3g). This means that only homogenous HAG complexes would be present at a high level in the single-mode oscillatory regime. One might ask, whether this is simply the result of higher mean levels of NF-$\kappa$B. Therefore, we normalised the concentration of the complexes by the mean NF-$\kappa$B concentration—when this ratio is large we will say the complexes are produced more efficiently. As seen in Fig. 3h, all complexes are produced more efficiently in the chaotic regime—even the homogenous HAG complexes. Another economical argument for the cell is that if only the complexes are of importance, then it is necessary to minimise the number of unused subunits. In Fig. 3i, we see the ratio between the average concentration of complexes to the concentration of unused subunits. This ratio too is largest in the chaotic regime for all complexes, except those made only from HAG proteins. Thus, a chaotically varying transcription factor not only up regulates LAGs, but also results in higher and more economical production of protein complexes composed of subunits from different genes.

**Chaos generates advantageous population heterogeneity**. We now consider how the dynamics of NF-$\kappa$B can affect a population of cells. In the following, we consider the deterministic NF-$\kappa$B system, and study a population of $N$ independent cells that are affected by the same oscillating TNF stimulus. In all simulations, cells have randomly distributed initial conditions, i.e., the NF-$\kappa$B oscillations in different cells are not initially synchronised. Within each cell, we will track one LAG and one HAG; parameters are chosen so that the two corresponding proteins have the same average protein level. In Fig. 4a–d we see that when NF-$\kappa$B is in a single-mode oscillatory state, the average level of both Protein 1 and Protein 2 is homogenous across the population, whereas if NF-$\kappa$B is mode-hopping then the distribution of protein levels across the populaiton is bimodal. In the chaotic regime, the distribution is broad and heterogenous for both proteins (Fig. 4e, f), but the LAG has on average a higher expression in this state (for TNF$_{Period} = 95$ min, we note a special tail, which is caused by the occurrence of some high-frequency oscillations).

Such heterogeneity in a cell population can provide a selective advantage when the population is exposed to some potentially lethal stresses. Imagine each cell in the population is exposed to two toxic drugs at concentrations $D_1$ and $D_2$. We assume that at each time step each cell is killed with

probability

$$\mathcal{P}_{Die} = \mathcal{P}_0 \left( \frac{D_1^h}{D_1^h + P_1^h} + \frac{D_2^h}{D_2^h + P_2^h} \right). \tag{7}$$

This describes a situation where the two proteins $P_1$ and $P_2$ are stress-responders that can help the cell survive stressed conditions. $\mathcal{P}_0$ represents the probability that the drugs kill in the absence of the protective proteins. We consider the case where $P_1$ is encoded by an HAG and $P_2$ by an LAG, both under control of NF-$\kappa$B.

First we consider the situation where only one of these drugs is present, shown in Fig. 4g, h. When only Drug 1 is added in a high amount, cells where NF-$\kappa$B is in a single-mode oscillating state will have a higher survival rate than cells where NF-$\kappa$B is mode-hopping or chaotic. This is what one would expect from Fig. 3, since HAG proteins are on average at higher levels in the single-mode oscillatory state. When only Drug 2 is added in a high amount, cells in chaotic states will have a slightly higher survival rate, but due to large fluctuations, these cells will also eventually die due to temporary low levels of Protein 2. Now we consider what will happen to the system if both drugs are added in a comparable amounts. We test four different patterns of adding the drugs (Fig. 4i–l) and find that the cells in the chaotic state will have significantly higher survival rate compared to the others. In the Supplementary Note 6, we provide some mathematical arguments for these results and here we also show tests of the robustness of these results, and here we found similar results as shown above (Supplementary Figure 3). From this we conclude that in the presence of multiple toxic drugs, a population of cells is better off having a large heterogeneity in gene expression and up regulating the LAGs and thus up regulating the product of genes. This is obtained in the chaotic regime for NF-$\kappa$B dynamics and this enhances the survival rate.

## Discussion

Transcription factors are known to have different dynamics, depending on external conditions, but how this may be exploited to differentially control downstream genes is not well understood. We have shown how dynamically varying transcription factors can differentially regulate genes based on an effective affinity that characterises the interaction between the gene and the transcription factor. In particular, we suggest that chaotic dynamics can produce differential control of high vs. LAGs, down regulating the former while simultaneously up regulating the latter. We show that this can be used not only to control single non-interacting genes, but also for upregulating specific complexes of proteins and generating useful heterogeneities in cell populations.

Our results are derived from a model of the NF-$\kappa$B system. Such models have been used to explain numerous experimentally observed features of NF-$\kappa$B oscillations[3,35], and therefore form a good basis for our exploration of the effects on downstream genes. Our model has already successfully predicted the existence of mode-hopping for a range of TNF amplitudes[21]. Since chaotic behaviour within overlapping Arnold tongues is such a fundamental feature of driven nonlinear oscillators[11,12,36], we are confident that NF-$\kappa$B driven by sufficiently large TNF amplitudes will exhibit deterministic chaos. However, an experimental realisation of our model[37,38] would necessarily be subject to various sources of noise and stochasticity, and it is not obvious that deterministically chaotic behaviour can be practically discerned in the presence of such fluctuations. Fortunately, many sophisticated methods exist that allow chaos to be distinguished from noise

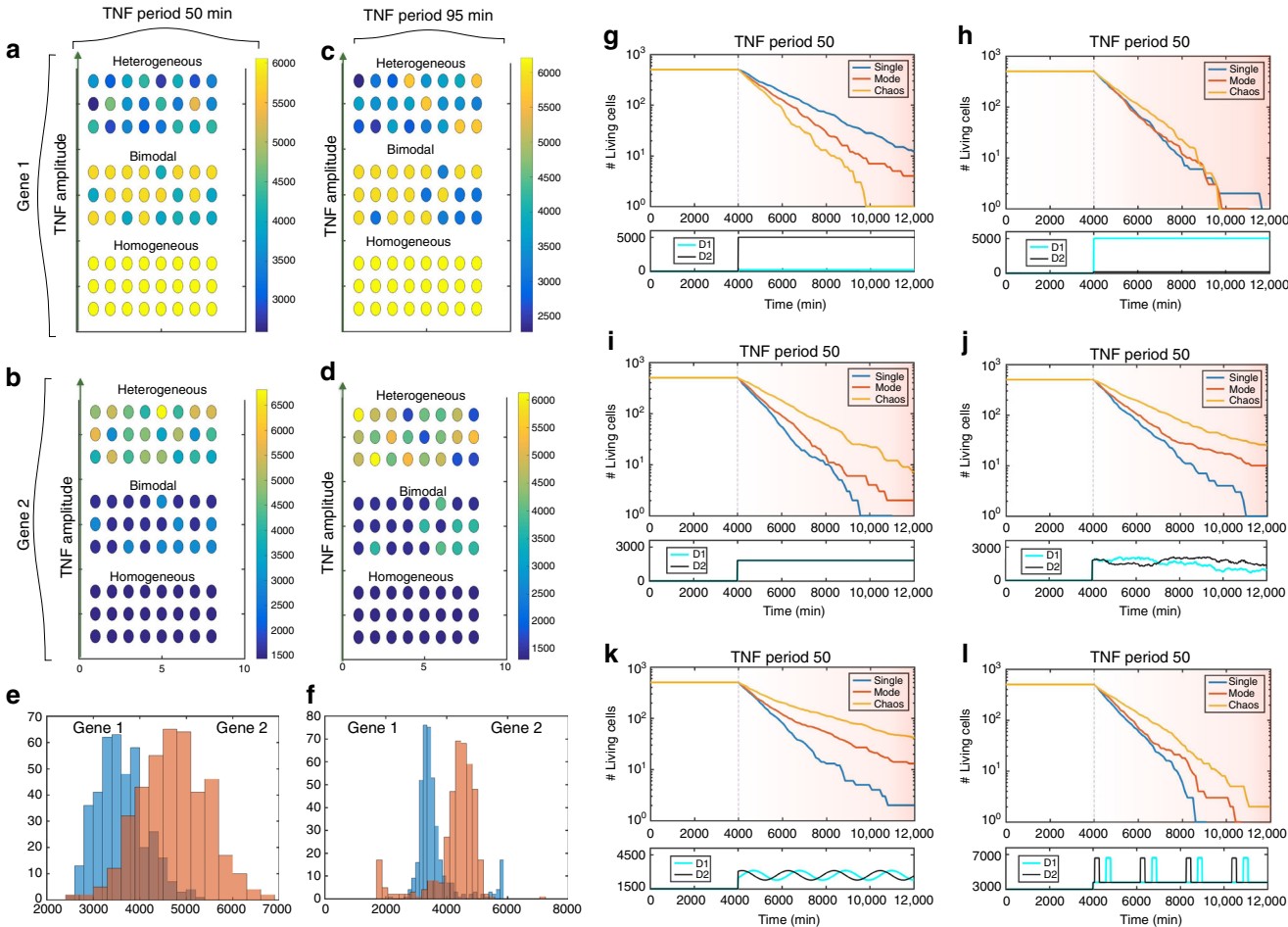

**Fig. 4** Population heterogeneity emerges from chaos. **a** Protein concentration from an HAG with $K = 1$, $h = 2$ and external TNF period 50 min. Bottom: The concentration corresponding to a single-mode oscillation; TNF amplitude: 0.04. Middle: The concentration corresponding to mode-hopping; TNF amplitude: 0.12. Top: The concentration corresponding to chaos; TNF amplitude: 0.36. **b** Protein concentration from an LAG with $K = 4.5$, $h = 4$. TNF period 50 min. TNF amplitudes are identical to those used in **a**. **c** Protein concentration from the HAG. TNF period 95 min. Bottom: The concentration corresponding to a single-mode oscillation; TNF amplitude: 0.1. Middle: The concentration corresponding to mode-hopping; TNF amplitude: 0.2. Top: The concentration corresponding to chaos; TNF amplitude: 0.4. **d** Protein concentration from the LAG. TNF period 95 min. TNF amplitudes are identical to those used in **c**. **e** Distribution of protein concentrations in the chaotic state (TNF period: 50 min, TNF amplitude: 0.36). **f** Distribution of protein concentrations in the chaotic state (TNF period: 95 min, TNF amplitude: 0.4). **g** Number of surviving cells vs. time (drug is added at $T = 4000$ min). $D1 = 6000$, $D2 = 0$. **h** Same as **g** with $D1 = 0$, $D2 = 6000$. **i** Same as **g** with $D1 = D2 = 3000$. **j** Same as **g** with $\dot{D}_{1+2} = \mathcal{N}(0, 100.0)$ and $D_{1+2}(0) = 3000$. The panel below shows a specific trajectory on this pattern. In general $D_1$ is above $D_2$ 50% of the times and vice versa. **k** Same as **g** with $D_{1+2}(t) = 3000 + 1500 \cdot \sin\left(\frac{t}{5000} + \Omega\right)$. **l** Same as **g** with $D_{1+2}(t) = 7000$ if $\sin\left(\frac{t}{5000} + \Omega\right) > 0.95$ and otherwise $D_{1+2}(t) = 3000$

without requiring unreasonably long time series; see for example refs. [39,40]. Once chaos is found in the NF-$\kappa$B system, the next step of testing whether HAGs respond differently from LAGs can be tackled using genes that have previously investigated in the regime where NF-$\kappa$B shows single-mode oscillations[1,8]. Since the expression level of some of these genes track NF-$\kappa$B oscillations closely, while others track the mean NF-$\kappa$B levels, it is likely that these genes already span a range of affinity values[4]. The robustness of our results to many parameter values suggests that these genes may be directly used to study the chaotic regime, without worrying too much about details, such as their maximal transcription/translation rates or the stabilities of the mRNA and proteins they encode.

Our model uses periodic variation of TNF to produce complex dynamics of NF-$\kappa$B. Uncovering conditions where TNF naturally varies periodically and thereby entrains the NF-$\kappa$B oscillations would add substantial weight to our results.

Oscillatory dynamics is believed to be of importance to several processes in the immune system[41] and there exists evidence that TNF does indeed vary in a pulsatile or periodic manner in some situations[42–45], as well as mathematical models that attempt to explain the underlying mechanisms[42,46], but it is unclear whether these natural oscillations entrain NF-$\kappa$B. The positive feedback between NF-$\kappa$B and TNF that has been hypothesised to produce travelling waves of TNF is perhaps the most promising scenario we are aware of where periodic TNF modulation may occur naturally[46].

Chaotic dynamics has thus far been underestimated as a means for controlling genes, perhaps because of its unpredictability. Our work shows that deterministic chaos potentially expands the toolbox available for single cells to control gene expression dynamically and specifically. We hope this will inspire theoretical and experimental exploration of the presence and utility of chaos in living cells.

## Methods

**Simulations**. All deterministic simulations were performed by numerically integrating the dynamical equations using the Runge–Kutta fourth-order method, and for optimisation reasons, some of the equations were simulated using Euler integration. Whenever Euler integration was used it was tested that it generated similar results as the Runge–Kutta fourth-order method. For all stochastic simulations of NF-kB dynamics, we used the Gillespie algorithm[23]. For noise in the external TNF oscillations we used Langevin simulations of the different oscillations.

**Regions of chaos**. To find the regions of parameter space that exhibit chaotic dynamics, we first computed the standard deviation in the NF-kB amplitudes from each time series, and found the parameter points at which this grew discontinuously, as we increased the TNF amplitude. Within these regions, we further tested for chaos by calculating the divergence of trajectories that started at almost identical initial points, using deterministic simulations. Parameter regions where such trajectories diverged exponentially were labelled as regions exhibiting chaos.

**Code availability**. All computer code is available upon request at heltberg@nbi.ku.dk or mhjensen@nbi.dk. The majority of scripts can also be found at https://github.com/Mathiasheltberg/ChaoticDynamicsInTranscriptionFactors.

## Data availability

All the data in this paper, was generated using deterministic and stochastic simulations. All scripts to generate the data are available upon request at heltberg@nbi.ku.dk or mhjensen@nbi.dk. The majority of scripts can also be found at https://github.com/Mathiasheltberg/ChaoticDynamicsInTranscriptionFactors.

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

## Acknowledgements

We are grateful Jordi Garcia-Ojalvo, Namiko Mitarai, Andrew Oates and Ala Trusina for valuable discussions. S.K. thanks the NCBS-TIFR and the Simons Foundation for funding. M.L.H. and M.H.J. acknowledge support from the Danish Council for Independent Research and Danish National Research Foundation through StemPhys Center of Excellence, grant number DNRF116.

## Author contributions

M.L.H., S.K. and M.H.J. developed the models. M.L.H. performed the numerical simulations. M.L.H., S.K. and M.H.J. wrote the paper.

## Additional information

**Competing interests:** The authors declare no competing interests.

