## [Peer Review File · Nature Communications]

Reviewers' comments:

Reviewer #1 (Remarks to the Author):

The manuscript by Heltberg et. al. presents a computational analysis of the NF-kappaB circuits and demonstrate that the chaotic dynamic generated in the system allows a degree of control of gene output and cell fate.

The authors consider their previously published model of the NF-kappaB system, which under a forced oscillatory regime coupled with intrinsic noise may lead to chaotic or state hopping behaviour (Fig.1). Authors show via simulations, that the chaotic behaviour might allow frequency-dependent differential regulation of low vs. high affinity genes (Fig.2). Following from this property, they demonstrate that one can differentially regulate multiprotein assembly (Fig. 3) and cell fate in multi-toxic environment (Fig.4). Therefore, authors argue that chaotic behaviour is a noise-independent way to modulate gene expression and cell state.

NF-kappaB is one of the key cellular signalling system, an established paradigm for how transcription factor dynamics regulates genes expression and cell fate. Previous analyses studied this system in response to continual or pulsatile stimulation regimes and argued that encoding is achieved both via intrinsic noise as well as extrinsic differences between individual cells. This work however demonstrates that the frequency-regulation can be achieved via chaos, which is potentially an important result with a generic applicability to other signalling systems. I believe that while the presented theoretical study demonstrates this principle reasonably well (see further comments below), I wonder however how biologically relevant and robust the findings are.

The key result is the differential control of the low and high affinity gene expression output between the single/multiple and chaotic states (Fig.2). As far as I understand the chaotic behaviour can be only obtained in the system by forcing the NF-kappaB responses with periodic TNF input (e.g. modelled via a sinusoid wave of different frequency/amplitude) in combination with the intrinsic noise. Previously, periodic forcing has been studied experimentally using in vitro microfluidic systems (work of Tay lab), however in vivo evidence that this is really a physiological stimulation/perturbation is rather unclear. The deterministic forcing seems like a very strong assumption, given potential heterogeneity of the in vivo tissue-level cytokine signalling. (Or in fact whether such pulses indeed happen at all). Therefore, it would be critical to test current predictions in the regime of "noisy forcing", e.g. with TNF input of variable amplitude and frequency. It would be critical to understand how robust the results are (e.g. differential gene expression, linearity of the output wrt frequency, Fig.2) while the level of the "forcing noise" changes. This would really allow testing whether the regime is biologically attainable. It might be that low level of "input noise" might mask the behaviour observed in the chaotic state.

Further question clarification of the simulation studies:

Fig 1C.

(i) What is the frequency of oscillations used in the simulations? Not clear, as subsequently altered in the paper.

(ii) Also not clear is the relative level of intrinsic noise in the system, i.e. how much the behaviour changes wrt the system size (V , Fig. 5) changes. In addition, previous works attributed heterogeneity due to bursty transcription (effectively assuming a stochastic telegraph models of transcription for activity of feedback genes)- not clear to me whether this effect is considered here and how it would affect the chaotic behaviour.

Fig. 2 (BTW. this figure is mislabelled in the text):

i) A-C. What is the frequency of oscillations?

ii) How the average of the production is defined? Ensemble average over some time, number of cells?

iii) Mode hopping regime, simulation appear not to be stably converging. Why? Is simply more simulations needed?

iv) Is the differential control true for different frequencies? What are the corresponding trajectories for different frequencies considered in D-F? Please for clarity show some other examples similar to plots A-C to establish how robust the behaviour is

v) I note small changes in the gene expression output, e.g. ~2-fold (~10% in Fig.2B). Given the noise in the system are the reported differences biologically measurable/relevant?

Figure 3:

i) Units of concentration: There is a 4-orders of magnitude difference between D and EF. This shouldn't be the case.

ii) G-H: the schematics of different complexes not clear, perhaps worth indicating red/blue ratios. Could you provide examples that are relevant for NF-kappaB signalling, that include different affinity genes/complexes (or in fact complexes of $N > 2$).

Figure 4:

i) analysis in G-L; not clear what is the frequency of TNF stimulation. In general, here I would like to see more analyses to understand how robust the presented effect is, e.g. in terms of how the frequency, parameters of gene regulation, etc affect survival rates. Also, what is not clear to me what is an intuitive explanation for the synergistic effect of the two genes.

Reviewer #2 (Remarks to the Author):

In this fascinating paper, the authors investigate theoretically and numerically how the dynamics of NF-KB, an important transcription factor, affects downstream genes. This is an important problem, because NF-KB is involved in the responses to many different types of stress and many different signals (ROS, tumor necrosis factor TNF, interleukin 1-beta,...) but activates only the cellular response which is appropriate for the stimulus. It is not currently understood how NF-KB manages to activate the specific downstream genes corresponding to this response. Since NF-KB displays a pulsatile dynamics, and can itself be driven by periodic pulses of TNF, it is suspected that some kind of dynamical encoding is at work, however the logic of this encoding has remained elusive so far.

More precisely, the authors note that when the NF-KB regulatory circuit is subjected to periodic TNF pulses of a fixed period but with different amplitudes, it can display very different types of dynamical behavior. At low amplitudes, the NF-KB temporal profile is periodic. At intermediate amplitudes, it displays a random alternation between two periodic waveforms, termed mode hopping. At higher amplitudes, NF-KB displays an irregular time evolution, known as deterministic chaos. This is a particular case of the fact that when an oscillator (such as the NF-KB module) is driven by an external cycle (such as the TNF signal), its response follows a complex pattern as the modulation period and amplitude are varied, known as an Arnold tongue. This universal mechanisms provides thus a simple way to generate very different NF-KB signals by simply tuning the TNF signal amplitude. This is not surprising for scientists familiar with dynamical system theory, but this fact has so far remained unappreciated in cellular biology and the authors are to be credited for their efforts to change this situation.

In this framework, the authors study how generic genes respond to these three different NF-KB

dynamical regimes depending on the affinity of NF-KB for their binding site and the cooperativity of the binding. This study leads to a series of surprising and far-reaching results, of high significance for cellular signaling.

First, the authors find that remarkably, high- and low-affinity genes have very contrasted responses to NF-KB dynamics. For high-affinity genes (HAG), expression is high for periodic behavior and low for deterministic chaos, and it is exactly the opposite for low-affinity genes (LAG). In the parameter regions corresponding to periodic and mode hopping dynamics, the expression is relatively constant, although at different levels. In the chaotic mode, expression varies almost linearly with TNF amplitude (decreasing for HAG, increasing for LAG). It follows that generating a chaotic NF-KB signal is a natural and simple way to have LAG expressed preferentially to HAG. In the chaotic regime, there is also the interesting observation that the response depends little on TNF period.

Then, the authors consider the problem of assembling a protein complex from several macromolecules. What if such a complex combines HAG and LAG proteins, whose synthesis requires different dynamical regimes for the driving signal? The authors find that the production of heterogeneous complexes is always favored by a wide margin when the NF-KB system operates in the chaotic regime, presumably because of the rich dynamic content of this regime. Moreover, they find that this makes the most efficient use of NF-KB factors and reduces the number of unused subunits.

Last but not least, the authors consider a population of cells and the expression levels of a HAG protein and a LAG protein. They find that in the periodic regime, these two proteins are expressed homogeneously in the population. In the mode hopping, where the NF-KB signal randomly alternates between two different waveforms, there is a bimodal distribution for each of the two proteins, each cell expressing the protein at either a high or a low level. Finally, when the NF-KB signal is chaotic, there is a complete spectrum of expression levels, producing a very heterogeneous population. This is a very important result because it is generally believed that heterogeneity in a population is produced by harnessing intrinsic noise, that is stochasticity and fluctuations in the biochemical reactions. The authors demonstrate here another simple mechanism to achieve this, which utilizes the randomness inherent to deterministic chaos.

The authors then provide an example of why generating a heterogeneous population may be advantageous. They assume that the two proteins are two antistress proteins which are designed to help the cells resisting to two types of stress. They find that when the two stresses are applied simultaneously, there is much less mortality in those populations where the two antistress proteins are expressed heterogeneously because the NF-KB is in the chaotic mode.

The only caveat here is that the authors show a snapshot of the population but since the cellular states presumably are dynamic, it would have been interesting to provide information on the time scale on which a given cell changes its state over the course of time.

These results are truly remarkable because there has been so far no clear evidence of the presence deterministic chaos in the cellular machinery, and there was no indication of how such complex dynamical regimes could be useful. I believe that this manuscript shows very convincingly that deterministic chaos may well be a fundamental ingredient of cellular signaling that has been overlooked so far, and that there are simple mechanisms exploiting the structure of Arnold tongues for generating chaotic signals in the NF-KB level. Indeed chaotic signals may be useful for (1) activating genes with low affinity for NF-KB, (2) producing efficiently heterogeneous protein complexes, (3) creating phenotypic diversity in a population of cells. This is simply fascinating and opens new research directions in cellular biology.

Of course, one may wonder how to validate experimentally the theoretical predictions of this manuscript. Fortunately, there exist sophisticated techniques for extracting signatures of chaos

from short signals (Amon and Lefranc, Phys. Rev. Lett. 92, 094101, 2004) which could be applied in biological systems.

Thus, I strongly recommend publication in a top journal like Nature Communications, however I believe that the following minor issues should be addressed first.

1) Although the results presented are outstanding, I have found that the manuscript could be much better written and that in many places it should be amended to be more precise and informative. The English also sounds strange sometimes. I really urge the authors to reread their manuscript with a critical eye, perhaps to have it proof-checked by someone else with a strong command of English and not familiar with the subject, and to rewrite it carefully, because this will affect directly the impact of the paper. Do not use concepts and terms before you have introduced them. Explain them in simple terms so that readers not expert in dynamical systems have a chance to understand.

2) The authors should not assume that the reader is familiar with the concept of chaos. Please define what you mean by that.

3) An example of the lack of precision: "In regions called Arnold Tongues". In regions of what? This is essential for the understanding of the reader. There are numerous sentences like that, I cannot list them all here. More generally the first paragraph should be more informative and discuss synchronization and entrainment, for example.

4) Inset : I would not oppose deterministic chaos with randomness, since chaos is effectively random at long times. It is only for short times that there is a difference between the two.

5) Fig.2 is a bit misleading in that the relative variation along the y axis is much smaller for B than for A and C. Please use scales so that there is for example a factor of two between the min and the max (appropriate for A and C). Please define the color code for figs DEF. Fig. 2 is referred to incorrectly as Fig. 3 many times.

6) In discussing Fig. 2, the authors should discuss why it is chaos that helps to express low-affinity genes, and not simply the high amplitude that correlates with chaos.

7) It is not surprising at all to find the pattern in the average of something driven by a chaotic signal, since chaotic signals have well-defined averages. Such useless remarks would be better removed.

8) I do not understand "this allows the existence of some genes that should only be expressed in rare cases".

9) I do not understand "genes from different families are stimulated" because of a spectrum of amplitudes, if it is the average that matters. Besides, there is still a rather well defined periodicity, it is only the amplitude that fluctuates from one period to the other.

10) "One potential advantage of the fluctuating dynamics ... could be the assembly of protein complexes" is absolutely not obvious to me. The authors should find a better transition, or detail their thinking.

11) why do the authors consider the same degradation rate for the two members of the protein complex. This looks like a severe restriction?

12) The authors should discuss how their findings could be validated experimentally, and in particular how it could be possible to evidence a chaotic dynamics in NF-KB. It would then be

appropriate to note that this could be done using techniques such as described by Amon et Lefranc.

Point-by-point response to the Reviewers' comments

We sincerely thank the two reviewers for their careful reading of and valuable comments on our manuscript. We have strived to make all the requested improvements, and we hope the revised manuscript is now considerably strengthened and suitable for publication in Nature Communications. Our specific responses are written below along with the original comments made by the reviewers.

Reviewer #1

I believe that while the presented theoretical study demonstrates this principle reasonably well (see further comments below), I wonder however how biologically relevant and robust the findings are.

The key result is the differential control of the low and high affinity gene expression output between the single/multiple and chaotic states (Fig.2). As far as I understand the chaotic behaviour can be only obtained in the system by forcing the NF-kappaB responses with periodic TNF input (e.g. modelled via a sinusoid wave of different frequency/amplitude) in combination with the intrinsic noise. Previously, periodic forcing has been studied experimentally using in vitro microfluidic systems (work of Tay lab), however in vivo evidence that this is really a physiological stimulation/perturbation is rather unclear. The deterministic forcing seems like a very strong assumption, given potential heterogeneity of the in vivo tissue-level cytokine signalling. (Or in fact whether such pulses indeed happen at all). Therefore, it would be critical to test current predictions in the regime of "noisy forcing", e.g. with TNF input of variable amplitude and frequency. It would be critical to understand how robust the results are (e.g. differential gene expression, linearity of the output wrt frequency, Fig.2) while the level of the "forcing noise" changes. This would really allow testing whether the regime is biologically attainable. It might be that low level of "input noise" might mask the behaviour observed in the chaotic state.

We thank the reviewer for pushing us to examine the biological relevance of our findings more thoroughly, especially with regard to our key result on differential control in Fig 2. We have found our results to be robust, as described below, and thus the reviewer's comments have helped us to considerably strengthen our paper. In the NF-kB system we suspect it is technically possible to obtain chaos without periodic forcing since it contains many of the nonlinearities present in other chaotic oscillators like the Rossler oscillator, but this is not something we have investigated. The easiest way to obtain chaos is in fact with periodic forcing **without** intrinsic noise (in the revised caption for Fig 1, we mention that the chaos seen in Fig 1C middle panel is from a deterministic simulation). However, we put in intrinsic noise precisely because we felt that a real biological scenario will always have intrinsic noise. To address the reviewer's second concern, we have added several references (Chan et al. 1999, Rayner et al. 2000, Ruohonen et al. 2005, Stark et al. 2007, Keller et al. 2009) to the revised manuscript's Discussion section that indicate the presence of periodic TNF modulation as well as pulses of TNF (akin to a noisy periodic TNF) under various conditions. Thus, we believe that there is a reasonable possibility that cells will encounter periodically varying TNF, albeit often with noise. Thus, as the reviewer suggested, it is crucial to test whether our result holds for noisy TNF forcing. We added various levels of such extrinsic noise to the sinusoidal TNF and found our results unchanged, as shown in the revised Fig. 2. Briefly, we find that such external noise has a significant effect only on the mode-hopping dynamics. It does not mask the transition into chaos and the large spectrum of frequencies and amplitudes that characterizes chaotic dynamics are still present in the presence of external noise. In the supplementary material, we show Fourier spectra (Fig. S2H) for the noisy waveforms we used (they consist of a deterministic sinusoid with noise added using a Langevin-like equation). The Fourier spectra show that we have added enough noise to substantially diminish the peak we should see from the sinusoid alone. We further tested the robustness of our results by varying the waveform of the periodic TNF

forcing (Fig. S2I), as well as varying several parameters (e.g., frequency, Hill coefficients, level of intrinsic noise, etc. – (Fig. S(A-J)). In all cases, we find that our results hold good, and we present these results in a new section in the revised manuscript entitled ‘*Robustness to variations in parameters, and intrinsic and extrinsic noise*’. The reviewer’s concerns also prompted us to develop a mathematical understanding of why we see the trends in Fig. 2, which we describe briefly in the main text and in more detail in the Supplementary section IV. Finally, we also tested the toxicity results against variation of parameters, and found that they too hold good across a broad range of parameter values. Together, we hope this shows both that chaos can occur in biologically relevant situations, and that it can allow for differential control of genes in biologically relevant ways. We reply to the additional comments of the reviewer point-by-point below:

Further question clarification of the simulation studies:

Fig 1C.

(i) What is the frequency of oscillations used in the simulations? Not clear, as subsequently altered in the paper.

Indeed, this was not clearly presented in the text. We have therefore explicitly included the frequency used in each figure caption (in all figures, not just Fig 1).

(ii) Also not clear is the relative level of intrinsic noise in the system, i.e. how much the behaviour changes wrt the system size (V , Fig. 5) changes. In addition, previous works attributed heterogeneity due to bursty transcription (effectively assuming a stochastic telegraph models of transcription for activity of feedback genes)- not clear to me whether this effect is considered here and how it would affect the chaotic behaviour.

Bursty transcription has not been explicitly included in our simulations, mainly because there is still considerable debate about the mechanism – two-state, three-state, and infinite-state models have been suggested. Based on previous studies of bursty transcription (e.g., Raj and Oudenaarden 2008), we believe its main effect in the context of our study would be to increase the intrinsic noise, which is also what reducing the effective volume does. Therefore, we repeated our simulations for multiple values of this effective volume, showing the results in the revised Fig. 2 and Fig S2. The previous Fig. 5, which showed the corresponding NF- κ B time series for these levels of noise, has been moved to the Supplement as Fig. S1. Overall, we find our results are robust to the variation of the intrinsic noise level caused by varying the effective volume. In particular, higher intrinsic noise (lower V) seems to have relatively little effect in the chaotic regime. Thus, both extrinsic noise (noise in TNF forcing) and intrinsic noise affect the system in similar ways, but leave our conclusions unaffected.

Fig. 2 (BTW. this figure is mislabelled in the text):

i) A-C. What is the frequency of oscillations?

The period of TNF oscillations here is 50 min. We have included this in the caption. And we have fixed the figure labelling in the text. We thank the reviewer for noticing these errors.

ii) How the average of the production is defined? Ensemble average over some time, number of cells?

We measure the time average over a very long time series. In principle, we should obtain the same answer from an ensemble average (i.e., averaging over many cells simulated in parallel at a given snapshot in time), however the time average is computationally much less intensive. However, to double check our results, we repeated our simulations for three additional cells, shown in the revised Fig 2. We find the average over time and over the four cells are similar, so our results do not change.

iii) Mode hopping regime, simulation appear not to be stably converging. Why? Is simply more simulations needed?

We thank the reviewer for pointing this out - this was exactly due to the fact that more simulations were needed. In the revised Fig. 2, we show that averaging over more simulations shows that the response converges to a relatively flat curve in the mode hopping region with, however, substantial fluctuations compared to the chaotic regime. This is not surprising because mode hopping arises from an underlying bistability where the dynamics can sometimes spend a very long time in one state before making the switch to the second state. Chaotic dynamics, in contrast, has much faster fluctuations and great variety in frequencies and amplitudes, so the average protein concentration reaches statistical equilibrium faster. We mention this in the revised text where we discuss Fig. 2.

iv) Is the differential control true for different frequencies? What are the corresponding trajectories for different frequencies considered in D-F? Please for clarity show some other examples similar to plots A-C to establish how robust the behaviour is.

The differential control is indeed true for different frequencies. In the revised Supplement, we show three additional response curves for TNF time periods 33 min, 70 min and 95 min. In addition, the heatmaps in Figs 2D-F also indicate that for all frequencies we have tested between 30 min and 110 min, we see a similar response, i.e., for high (low) K the average production decreases (increases) as TNF amplitude increases.

v) I note small changes in the gene expression output, e.g. ~2-fold (~10% in Fig.2B). Given the noise in the system are the reported differences biologically measurable/relevant?

We agree with the reviewer that the fold-changes in Fig 2B (~10%) are not likely to be biologically significant. However, the fold changes in Figs 2A and 2C are, we believe, both measurable and biologically significant. Tay and Kellogg, 2015, and Hoffmann et al., 2002, who have looked at genes downstream of NF κ B, have reported many known targets that show a similar fold-change. Thus, our conclusion is that varying the dynamics of NF- κ B would make a difference to genes with low and high K, but there would be a range of intermediate K where the genes would be relatively unaffected. We find this an interesting point - as useful as it is to have genes whose expression is affected significantly by NF- κ B, it may be equally useful to have ways to “buffer” certain genes from the chaotic fluctuations of NF- κ B when required, and our results show that both can be achieved. We have commented on this point in the main text where we discuss Fig 2, and in order to make this clearer we have also made the y-axes scale of Fig. 2A-C more comparable. In addition, we now normalize the average production to the unperturbed case (where TNF is flat, not periodic) in order to allow a sensible comparison between Figs. 2A, 2B and 2C.

Figure 3:

i) Units of concentration: There is a 4-orders of magnitude difference between D and EF. This shouldn't be the case.

This is a mistake in the figure and we have corrected this. In order to make the figure clearer and more informative, in the revised figure we have also used the dimensionless ratio between protein levels in the chaotic and single-mode state.

ii) G-H: the schematics of different complexes not clear, perhaps worth indicating red/blue ratios. Could you provide examples that are relevant for NF- κ B signalling, that include different affinity genes/complexes (or in fact complexes of $N>2$).

We have labelled some of the complexes in Fig. 3(G-I) with their blue:total ratios – we hope this improves the readability of the figure while highlighting the main result which is that heterogeneous complexes are upregulated in the chaotic state. To avoid clutter we did not label all the complexes

shown, however in Supplementary material we have included tables showing the entire data-set plotted in Fig. 3(G-I), with all ratios. Regarding examples of NF- κ B regulated protein complexes, Tieri et al., 2012, have attempted to map the 'NF- κ B interactome'. Within their data we found a dataset composed of 384 genes that are known to be downstream of NF- κ B (they call this the DG dataset). The protein-protein interaction network for these 384 proteins contains 572 links, i.e. there are 572 pairs which form complexes and also there are several proteins which have more than one partner indicating some possibility of multi-protein complexes. Thus far, the functional role of these complexes has not been established but given the ubiquity of multi-protein complexes in eukaryotic regulation we believe this data provides a good motivation for our theoretical study of complexes downstream of NF- κ B. In addition, there is some evidence of NF- κ B regulated complexes which play a role in autophagy. NF- κ B appears to influence autophagy via multiple pathways which may interact – one possibility is Beclin 1 and A20, both proteins are regulated by NF- κ B, and A20 inhibits ubiquitination of Beclin 1 thus repressing autophagy (Trocoli et al. 2011). In the revised manuscript we mention all this in the results section where we investigate complexes.

Figure 4:

i) analysis in G-L; not clear what is the frequency of TNF stimulation. In general, here I would like to see more analyses to understand how robust the presented effect is, e.g. in terms of how the frequency, parameters of gene regulation, etc affect survival rates. Also, what is not clear to me what is an intuitive explanation for the synergistic effect of the two genes.

We thank the reviewer for pushing us to test the robustness of this result to variation in parameters. In the revised manuscript, we have included a figure in the supplementary material, Fig. S3, where we find similar results for different frequencies, different amplitudes and different hill coefficients. Regarding the reason for this synergistic effect, we believe it stems from the fact that the toxicity is inversely affected by the product of the concentrations of the two proteins (the mathematical argument for this is included in Supplementary section VI). As shown in the new Supplementary Fig. S3, as we move from the single mode state to the chaotic state, the sum of protein concentrations is unchanged on average, but the product increases significantly. Thus, the toxicity decreases overall. The fact that the product of concentrations increases follows directly from our main result in Fig. 2. Thus, we are able to connect the behaviour seen in Fig. 4 to our result from Fig. 2. We hope this will serve as the intuitive explanation the reviewer desired. We have mentioned this briefly in the main text and in more detail in section VI of the Supplement.

Reviewer #2:

Thus, I strongly recommend publication in a top journal like Nature Communications, however I believe that the following minor issues should be addressed first.

We thank the reviewer for the kind comments. We are very pleased that he/she found the paper fascinating and worthy of publication in Nature Communications. We indeed hope our work will open up a whole new set of research questions for both experimentalists and theorists interested in dynamic ways of regulating genes. We are also grateful that the reviewer points to ways to make the presentation of our findings more understandable and accessible to a broad audience, and we hope our revisions have achieved this. Below we address each of the reviewer's concerns point-by-point.

1) Although the results presented are outstanding, I have found that the manuscript could be much better written and that in many places it should be amended to be more precise and informative. The English also sounds strange sometimes. I really urge the authors to reread their manuscript with a critical eye, perhaps to have it proof-checked by someone else with a strong command of English and not familiar with the subject, and to rewrite it carefully, because this will affect directly the impact of the paper. Do not use concepts and terms before you have introduced them. Explain them in simple terms so that readers not expert in dynamical systems have a chance to

understand.

We completely agree with the reviewer, and in the revised manuscript we have consulted with native speakers of English to reword many parts to make the arguments clearer and more accessible to a broader audience. We have tried to be more careful about our use of terminology, and in particular have defined nonlinear dynamics terms before using them so that it is more understandable to a wider audience. This has resulted in considerable changes to the text – mainly in the Introduction, Discussion, and first section of Results – but the logic of the arguments has not altered, the changes are only in the interest of increasing readability. In addition, we have left the figures untouched (except for the additional data in the revised Fig. 2 to answer reviewer concerns). We hope both reviewers will find the new manuscript to be much clearer.

2) The authors should not assume that the reader is familiar with the concept of chaos. Please define what you mean by that.

We have tried to describe this better in the box, which is now titled "What is Chaos?".

3) An example of the lack of precision: "In regions called Arnold Tongues". In regions of what? This is essential for the understanding of the reader. There are numerous sentences like that, I cannot list them all here. More generally the first paragraph should be more informative and discuss synchronization and entrainment, for example.

We now define Arnold tongues, and similarly unfamiliar terms, before using them. We have also rewritten the first paragraph and other parts of the introduction to familiarize readers with concepts that we use later in the paper.

4) Inset : I would not oppose deterministic chaos with randomness, since chaos is effectively random at long times. It is only for short times that there is a difference between the two.

We thank the reviewer for pointing this out. It was not our intention to oppose chaos with randomness - we have rewritten the text in the box, and wherever needed in the main text, in order to make this point clear.

5) Fig.2 is a bit misleading in that the relative variation along the y axis is much smaller for B than for A and C. Please use scales so that there is for example a factor of two between the min and the max (appropriate for A and C). Please define the color code for figs DEF. Fig. 2 is referred to incorrectly as Fig. 3 many times.

We agree with the reviewer and have re-scaled the y-axes to be more comparable, while still leaving the trends visible in the response curves. We have also normalized the data with respect to the unperturbed case (with flat TNF) in order to allow an easier comparison between Fig. 2A, B and C. We have added information about the colour codes and corrected the references to the figure in the main text. We thank the reviewer for pointing out these errors.

6) In discussing Fig. 2, the authors should discuss why it is chaos that helps to express low-affinity genes, and not simply the high amplitude that correlates with chaos.

We now have a more analytic understanding of this result, which is explained briefly in the text and in more detail in Supplementary section IV. To summarize: the distribution of NF- κ B levels over a long time-series changes in a systematic manner as one moves from the single-mode to chaos, as well as when one increases the TNF amplitude within the chaotic regime. Essentially, the deeper one moves into chaos, the broader the tails of the distribution on both the left (small NF- κ B) and right (large NF- κ B). We argue that in the limit of small K (high affinity genes), the average gene expression is dominated by the left tail, while in the limit of large K (low affinity genes) the average expression is dominated by the right tail. This is what leads to the opposite behaviour of these two

types of genes as a function of TNF amplitude. We note that in both cases, the NF- κ B distribution changes less in its mean/median and more in the tails. Thus, we believe the differential control of low and high affinity genes arise not because of changes in the average level of NF κ B due to changing TNF forcing amplitude, but due to subtler changes in the distribution. However, it is fair to say that the broadening of the NF- κ B distribution as one goes deeper into chaos (which can be thought of as an increase in the NF- κ B amplitude) is the underlying cause for the differential control of low vs high affinity genes. We hope the reviewer finds this argument interesting and worthy of inclusion in the revised manuscript.

7) *It is not surprising at all to find the pattern in the average of something driven by a chaotic signal, since chaotic signals have well-defined averages. Such useless remarks would be better removed.*

We have removed this from the text, along with other remarks that we thought were not adding any useful information for the reader.

8) *I do not understand “this allows the existence of some genes that should only be expressed in rare cases”.*

We have removed this confusing sentence, and rewritten this part of the text to make the overall argument more clear.

9) I do not understand “genes from different families are stimulated” because of a spectrum of amplitudes, if it is the average that matters. Besides, there is still a rather well defined periodicity, it is only the amplitude that fluctuates from one period to the other.

We have removed this confusing sentence.

10) *“One potential advantage of the fluctuating dynamics ... could be the assembly of protein complexes” is absolutely not obvious to me. The authors should find a better transition, or detail their thinking.*

We have rewritten this confusing portion. We hope the reviewer finds the new formulation to be clearer.

11) *why do the authors consider the same degradation rate for the two members of the protein complex. This looks like a severe restriction?*

This is a very good point by the reviewer, and in reality one could easily believe that the degradation rates could differ. However, as this is to our knowledge the first attempt to investigate differential control of such heterogenous complexes, we made this assumption entirely in the interest of simplicity. We hope that despite restricting the parameter space, the reviewers will still find our results intriguing and indicative for future work, which certainly should explore asymmetric degradation. We do mention this more explicitly in the Results section concerning complexes of the revised manuscript.

12) *The authors should discuss how their findings could be validated experimentally, and in particular how it could be possible to evidence a chaotic dynamics in NF- κ B. It would then be appropriate to note that this could be done using techniques such as described by Amon et al.*

This is a crucial point and we thank the reviewer for bringing to our notice the Amon et al paper, which provides a very useful and practical way of detecting chaos with the kinds of time series that experimentalists can feasibly generate for the NF- κ B system. We have added a paragraph to the Discussion section which suggests ways to validate our findings by combining theoretical methods

like those in Amon et al with previously attempted manipulation of TNF in studies such as [Piehler et al. 2017, Dettinger et al. 2018].

References:

Piehler, A., Ghorashian, N., Zhang, C., Tay, S.
Universal signal generator for dynamic cell stimulation.
Lab on a Chip, 17(13), 2218-2224 (2017).

Dettinger, P., Frank, T., Etzrodt, M., Ahmed, N., Reimann, A., Trenzinger, C., ... Tay, S.
Automated Microfluidic System for Dynamic Stimulation and Tracking of Single Cells.
Analytical Chemistry (2018).

Amon, A., Lefranc, M.
Topological signature of deterministic chaos in short nonstationary signals from an optical parametric oscillator.
Physical review letters, 92(9), 094101 (2004).

Stark, J., Chan, C., & George, A. J.
Oscillations in the immune system.
Immunological reviews, 216(1), 213-231 (2007).

Chan, C. C., Stark, J., & George, A. J.
Analysis of cytokine dynamics in corneal allograft rejection.
Proceedings of the Royal Society of London B: Biological Sciences, 266(1434), 2217-2223 (1999).

Rayner, S. A., King, W. J., Comer, R. M., Isaacs, J. D., Hale, G., George, A. J. T., & Larkin, D. F. P.
Local bioactive tumour necrosis factor (TNF) in corneal allotransplantation. *Clinical & Experimental Immunology*, 122(1), 109-116 (2000).

Ruohonen, S., Khademi, M., Jagodic, M., Taskinen, H. S., Olsson, T., & R oytt a, M.
Cytokine responses during chronic denervation. *Journal of neuroinflammation*, 2(1), 26 (2005).

Keller, M., Mazuch, J., Abraham, U., Eom, G. D., Herzog, E. D., Volk, H. D., ... & Maier, B.
A circadian clock in macrophages controls inflammatory immune responses. *Proceedings of the National Academy of Sciences*, 106(50), 21407-21412 (2009).

Raj A, van Oudenaarden A.
Stochastic gene expression and its consequences.
Cell. 2008;135(2):216-226. doi:10.1016/j.cell.2008.09.050.

Hoffmann, A., Levchenko, A., Scott, M. L., & Baltimore, D.
The I κ B-NF- κ B signaling module: temporal control and selective gene activation. *Science*, 298(5596), 1241-1245 (2002).

Kellogg, R. A., & Tay, S.
Noise facilitates transcriptional control under dynamic inputs.
Cell, 160(3), 381-392 (2015).

Tieri, P., Termanini, A., Bellavista, E., Salvioli, S., Capri, M., & Franceschi, C. Charting the NF- κ B pathway interactome map.
PloS one, 7(3), e32678 (2012).

Trocoli, A., & Djavaheri-Mergny, M.

The complex interplay between autophagy and NF- κ B signaling pathways in cancer cells.
American journal of cancer research, 1(5), 629 (2011).

REVIEWERS' COMMENTS:

Reviewer #1 (Remarks to the Author):

The revised manuscript fully addresses the points raised in the review. I strongly recommend the paper for publication.

Reviewer #2 (Remarks to the Author):

I find that the authors have satisfactorily addressed all the issues raised in my previous report as well as by the other referee. I particularly appreciate the improved clarity of the manuscript, the discussion of how the main findings are robust to changes in parameters as well as to extrinsic or intrinsic noise, as well as the argument explaining why low-affinity genes respond to a chaotic waveform, among other improvements. This strengthens the manuscript substantially and makes the central claims more plausible.

As I commented in my previous report, this is a landmark paper regarding how complex transcription factor dynamics may affect gene regulation. It describes many remarkable and non-intuitive effects which may well have been harnessed by nature. It is well argued and the evidence is convincing. Moreover, this is perhaps the first time where a plausible use of chaotic dynamics in cellular biology is proposed.

Thus I wholeheartedly recommend this paper for publication in Nature Communications.

Reply to Reviewer #1

Reviewer #1

The revised manuscript fully addresses the points raised in the review. I strongly recommend the paper for publication.

Our reply:

We thank the reviewer for the excellent points that has been raised. We feel that this reviewing process has improved the manuscript very much and in answering the questions regarding robustness, we have strengthened the conclusions enormously.

Reply to Reviewer #2

Reviewer #2

I find that the authors have satisfactorily addressed all the issues raised in my previous report as well as by the other referee. I particularly appreciate the improved clarity of the manuscript, the discussion of how the main findings are robust to changes in parameters as well as to extrinsic or intrinsic noise, as well as the argument explaining why low-affinity genes respond to a chaotic waveform, among other improvements. This strengthens the manuscript substantially and makes the central claims more plausible. As I commented in my previous report, this is a landmark paper regarding how complex transcription factor dynamics may affect gene regulation. It describes many remarkable and non-intuitive effects which may well have been harnessed by nature. It is well argued and the evidence is convincing. Moreover, this is perhaps the first time where a plausible use of chaotic dynamics in cellular biology is proposed. Thus I wholeheartedly recommend this paper for publication in Nature Communications.

Our reply:

We thank the reviewer sincerely for the nice words and for raising some excellent questions. We feel that the critical proof reading of the manuscript has led to a serious improvement and we feel that the conclusions of this work stand much clearer with this present version. Furthermore we are happy about the point on how to relate this work to future experiments and give a thorough way on how to look for chaos in experimental trajectories. Therefore we feel that the reviewing process has improved the paper substantially.